# Assessing the Accuracy of GEDI Data for Canopy Height and Aboveground Biomass Estimates in Mediterranean Forests

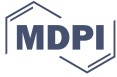

**Iván Dorado-Roda** [1], **Adrián Pascual** [2], **Sergio Godinho** [3,4], **Carlos A. Silva** [5,6,7], **Brigite Botequim** [8], **Pablo Rodríguez-Gonzálvez** [1], **Eduardo González-Ferreiro** [1] and **Juan Guerra-Hernández** [8,9,*]

1. Departamento de Tecnología Minera, Topografía y de Estructuras, Escuela Superior y Técnica de Ingenieros de Minas y Escuela de Ingeniería Agraria y Forestal, Universidad de León, Av. de Astorga s/n, Campus de Ponferrada, 24401 Ponferrada, Spain; idorar00@estudiantes.unileon.es (I.D.-R.); p.rodriguez@unileon.es (P.R.-G.); egonf@unileon.es (E.G.-F.)
2. Center for Global Discovery and Conservation Science, Arizona State University, Hilo, HA 96720, USA; apascua6@asu.edu
3. EaRSLab—Earth Remote Sensing Laboratory, University of Évora, 7000-671 Évora, Portugal; sgodinho@uevora.pt
4. Institute of Earth Sciences (ICT), Universidade de Évora, Rua Romão Ramalho, 59, 7002-554 Évora, Portugal
5. School of Forest Resources and Conservation, University of Florida, P.O. Box 110410, Gainesville, FL 32611, USA; c.silva@ufl.edu
6. Department of Geographical Sciences, University of Maryland, College Park, MD 20740, USA
7. Biosciences Laboratory, NASA Goddard Space Flight Center, Greenbelt, MD 20707, USA
8. Forest Research Centre, Instituto Superior de Agronomia (ISA), School of Agriculture, University of Lisbon, Tapada da Ajuda, 1349-017 Lisboa, Portugal; bbotequim@isa.ulisboa.pt
9. Centro de Iniciativas Empresariais, Fundación CEL, O Palomar s/n, 27004 Lugo, Spain
* Correspondence: juan.guerra@3edata.com; Tel.: +351-919-878-010

**Abstract:** Global Ecosystem Dynamics Investigation (GEDI) satellite mission is expanding the spatial bounds and temporal resolution of large-scale mapping applications. Integrating the recent GEDI data into Airborne Laser Scanning (ALS)-derived estimations represents a global opportunity to update and extend forest models based on area based approaches (ABA) considering temporal and spatial dynamics. This study evaluates the effect of combining ALS-based aboveground biomass (AGB) estimates with GEDI-derived models by using temporally coincident datasets. A gradient of forest ecosystems, distributed through 21,766 km$^2$ in the province of Badajoz (Spain), with different species and structural complexity, was used to: (*i*) assess the accuracy of GEDI canopy height in five Mediterranean Ecosystems and (*ii*) develop GEDI-based AGB models when using ALS-derived AGB estimates at GEDI footprint level. In terms of Pearson's correlation (r) and rRMSE, the agreement between ALS and GEDI statistics on canopy height was stronger in the denser and homogeneous coniferous forest of *P. pinaster* and *P. pinea* than in sparse *Quercus*-dominated forests. The GEDI-derived AGB models using relative height and vertical canopy metrics yielded a model efficiency (Mef) ranging from 0.31 to 0.46, with a RMSE ranging from 14.13 to 32.16 Mg/ha and rRMSE from 38.17 to 84.74%, at GEDI footprint level by forest type. The impact of forest structure confirmed previous studies achievements, since GEDI data showed higher uncertainty in highly multilayered forests. In general, GEDI-derived models (GEDI-like Level4A) underestimated AGB over lower and higher ALS-derived AGB intervals. The proposed models could also be used to monitor biomass stocks at large-scale by using GEDI footprint level in Mediterranean areas, especially in remote and hard-to-reach areas for forest inventory. The findings from this study serve to provide an initial evaluation of GEDI data for estimating AGB in Mediterranean forest.

**Keywords:** aboveground carbon; forest monitoring; spaceborne LiDAR; data fusion

## 1. Introduction

Based on the general FAO definition of forests, there were an estimated 88 million ha of forest area in Mediterranean countries in 2015, occupying the 10.04% of the total area of

these countries and representing 2.20% of the world's total forest area [1]. In 2015, these forests stored 5066 billion tons of carbon, equivalent to 1.7% of global forest carbon [1]. Additionally, in some cases, this forest definition could exclude important vegetative formations, such as most open oak woodlands of *Quercus* species (e.g., Spanish *Dehesas* and Portuguese *Montados*) [2], that would be contributing significantly to carbon storage. Effective large-scale monitoring of Mediterranean forest ecosystem therefore has a critical role for adapting to climate change [3,4]. The recommendation of the Intergovernmental Panel on Climate Change (IPCC) is to use a combination of Earth observation (EO) data and field-based inventories to estimate the forest area, carbon stocks, and changes [5]. Nationwide surveys in the form of a National Forest Inventory (NFI) have contributed by means of extensive fieldwork campaigns to monitor the dynamics of the Land Use, Land Use Change and Forestry (LULUCF) sector [6]. The monitoring of aboveground carbon represents an operational challenge under NFI sampling designs, and it may not provide reliable local estimates, for instance, at sub-regional and national level [7]. The support from remote sensing data (RS) has contributed to a better understanding of the spatial drivers of variation in LULUCF global flagship indicators such as forest aboveground biomass (AGB) [5]. In fact, Eggleston et al. (2006) [8] have listed AGB as one of the most important carbon pools by representing around 30% of the total terrestrial ecosystem carbon pool [9]. Therefore, accurately mapping and monitoring the spatio-temporal distribution of AGB at local, regional, and global scales is a crucial step towards an effective carbon stocks quantification and consequently a better climate change mitigation plan. Spatially continuous information on forest 3D structure is essential to estimate AGB distribution [10]. During the past three decades, airborne laser scanning (ALS) has become an established solid method for accurately mapping key indicators for Mediterranean forests (e.g., canopy height [11], forest inventory variables [12], aboveground carbon [13], and canopy fuel characteristics [14]) at high spatial resolution and in a relatively short time compared to conventional methods. The high costs of surveying at local/regional scale using ALS technology can result in low temporal resolution of AGB estimates, narrowing the potential of ALS technology at vectorizing the estimation of AGB maps in many countries. The situation has changed recently once global spaceborne laser scanning missions have turned operational and accessible [15,16].

The Ice, Cloud and land Elevation Satellite (ICESat-2) [17] and the Global Ecosystem Dynamics Investigation (GEDI) [18] missions are continuously scanning over the Earth's land surfaces since 2018 and 2019, respectively. The Advanced Topographic Laser Altimeter System (ATLAS) on-board the ICESat-2 satellite and GEDI mission on-board the International Space Station (ISS) are currently generating dense along-track LiDAR information at large spatial coverages and can be used to support carbon monitoring and fuel mapping estimations at global level. GEDI and ICESat2 mission depends on worldwide, crowd-sourced in situ field inventories, and ALS datasets to develop representative pre-launch calibration equations for predicting AGB across the National Aeronautics and Space Administration (NASA) observation domain [19–21]. GEDI mission transports a full-waveform (FW) Light Detection and Ranging (LiDAR) instrument to monitor tropical and temperate forests ecosystems. A robust post-launch validation is therefore needed to assess the accuracy of the spaceborne LiDAR sensor, especially due to the difficulties in differentiating AGB and characterizing the vegetation structure under sparse forest cover, which is characteristic of, e.g., *Quercus spp.* Mediterranean forests [22]. Spanish country-wide low-density ALS data has been successfully used for estimating forestry attributes in Mediterranean forests using the area based approach (ABA) (e.g., [9,18]). In the case of Extremadura region (Spain), well-georeferenced Fourth Spanish National Forest Inventory (SNFI-4) plots and low-density ALS data have greatly contributed to improve large-scale estimates of biophysical forest attributes [3,4,13,23]. The efficiency of the ABA method deeply relies on the temporal and spatial alignment between field measurements and laser statistics. Since, top-height vegetation maps derived from GEDI technology are being developed nowadays worldwide, across a gradient of complex environments [10,24–28],

many researchers could consider using GEDI data as the only source on which to rely for forest structure estimation, in the absence of multi-temporal ALS in the near future. Hence, developing GEDI-based AGB models from Mediterranean areas using existing ALS-based AGB estimates towards the domain of new spaceborne datasets (specifically to those designed to map ecosystem structure observation, i.e., GEDI, ICESat2, NASA's NISAR, ESA's BIOMASS) is a relevant exercise, since satellite-derived data offers unique opportunities to produce large-scale forest estimates on 3D forest structure.

Several studies have assessed the accuracy of GEDI and ICESat-2-derived canopy heights [10,24,25,28] and AGB estimates [15,20,21,29] from different ecosystems around the world. However, canopy height was validated using non coincident temporal or simulated ALS data [10,24,25,28] and simulated GEDI and ICESat-2-data to calibrate global AGB models [15,20,21,29]. Hence, it is important, first, to evaluate the accuracies of post-launch GEDI data and products, since they might differ from the accuracies of the simulated GEDI data, and secondly, to calibrate/validate the local and regional spaceborne LiDAR-AGB model. The dynamics of forest ecosystems are especially challenging in sparse ecosystems for which the interpretation of laser echoes remains more uncertain than in full-cover conditions, where the upper canopy layer more uniformly captures the energy from laser beams. The case of Mediterranean forests could provide valuable insights on how forest horizontal complexity determines the performance of GEDI-based estimates when compared to ALS-based estimation. The study analyzes a gradient of forest ecosystems from sparse to dense forest cover conditions on which to show the performance of ABA models when applied using recent ALS surveys (2018–2019) and recent GEDI scanning shots (2019). The spatial and temporal co-registration between ALS and GEDI shots were specifically considered in this investigation. To the best of our knowledge, no studies have been conducted in Mediterranean areas focused on developing AGB models using ongoing satellite LiDAR missions. This is a crucial step to better understand forest AGB distribution and spatial changes in terrestrial carbon fluxes. Therefore, the main goal of this work is to assess the main capabilities of the GEDI sensor for estimating canopy height and AGB over five different Mediterranean forest ecosystems in south-west Spain. To fulfill this goal, two specific objectives were defined: (1) assess the accuracy of GEDI-derived canopy height, and (2) quantify the performance of the GEDI-derived models based on canopy metrics (height and cover) in predicting AGB. To do that, the ALS-derived canopy height and AGB computed for the study area were used as reference data.

## 2. Materials and Methods

### 2.1. Study Area

The study area was located in the province of Badajoz (region of Extremadura, south-west of Spain). Badajoz is the biggest province in Spain covering 21,766 km$^2$ (Figure 1). The Spanish Forest Map (SFM) and the sampling design of the SNFI-4 were used in this research to cover a wide range of Mediterranean types of forests. The last version of the SFM at 25-m resolution was used to select the forest areas corresponding to the most representative species in Extremadura. We selected five forest ecosystems: (*i*) *Dehesas*: agro-forestry–pastoral ecosystem that contains scattered tree cover (60–100 trees per ha) dominated by even-aged old-growth evergreen oaks (*Quercus spp.*) usually with an absence of natural regeneration due to the presence of pastures and agricultural fields as undercover; (*ii*) *Encinar*: uneven-aged sparse oak forest (*Quercus ilex* subsp. ballota (Desf.) Samp); (*iii*) *Alcornocales*: even-aged multilayered forest dominated by the cork oak (*Quercus suber* L.); (*iv*) *Pinaster*: even-aged forests of *Pinus pinaster* subsp. mesogeensis Aiton; and (*v*) *Pinea*: even-aged forests of *Pinus pinea*. According to the SFM, the analyzed forest stands cover a total 822,623 ha distributed as follows: 664,529.97 ha (*Dehesas*), 98,950.11 ha (*Encinares*), 14,482.87 ha (*Pinaster*), 17,052.81 ha (*Alcornocales*), and 27,610.21 ha (*Pinea*).

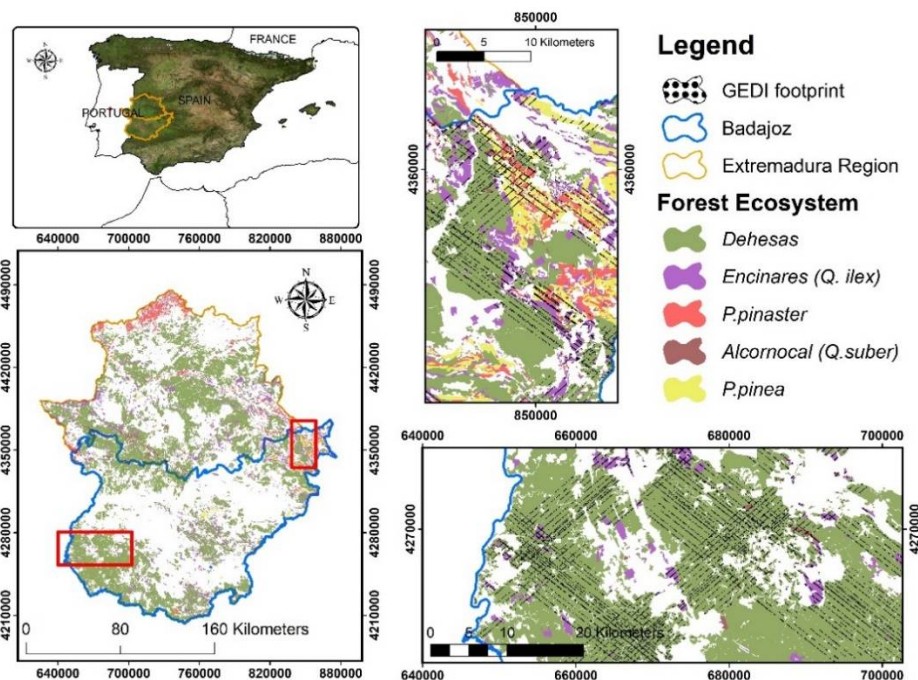

**Figure 1.** Boundary of the 'Badajoz Province' forest study site (blue line) and the locations of GEDI shots inside the different Mediterranean forest types. The red rectangles zooms represent the distribution of GEDI orbital tracks across the different Forest Ecosystems.

### 2.2. Airborne Laser Scanning Acquisition and Processing

ALS campaigns can be considered temporally coincident with the GEDI track, since there was a maximum of 1 year time interval between ALS acquisition date and the set of the analyzed GEDI full waveform (FW) beam footprint. Two sets of ALS point clouds were processed in this study: (*i*) Extremadura North (EXT-N, collected during the period of October 2018 to March 2019) and (*ii*) Extremadura South (EXT-S, collected during the period of October 2018 to July 2019). Both data sets correspond to the second round of countrywide ALS measurements, which are publicly available in Spain through the PNOA project (*Plan Nacional de Ortofotografía Aérea*). Squared ALS blocks of 2-km side, covering the whole region of Extremadura were obtained from the CNIG ("*Centro Nacional de Infomación Geográfica*" http://centrodedescargas.cnig.es/CentroDescargas/index.jsp, accessed on 1 June 2020) to cover the province of Badajoz. The scanning sensors involved in collecting the ALS data in the study area were a RIEGL LMS-Q1560 for the EXT-N dataset and a LEICA ALS80 for the EXT-S dataset. The nominal laser pulse density varied between 2 points m$^{-2}$ in the EXT-N and 1 point m$^{-2}$ in the EXT-S. The vertical accuracy of the scanning survey was 0.15 m for both ALS datasets.

The processing workflow comprised the following steps: Firstly, *thindata* command implemented in FUSION software [30] was used to reduce the nominal pulse density to 1 point m$^{-2}$ in order to homogenize the results from both datasets (EXT-N, EXT-S). Secondly, the ALS data sets were processed using the LAStools software [31]. A detailed description of the software parametrization and processing workflow is provided in Pascual et al., 2020 [4]. Briefly, *lasheight* were used to normalize the classified point cloud of ALS echoes. The *lascanopy* command was used to extract the metrics from the ALS normalized point cloud using a buffer of 12.5 m of radius for the center of each GEDI footprint (Table 1). Finally, the above-ground height of ALS echoes was used to distinguish tree canopies (echoes above 2 m) and the shrub layer (echoes below 2 m) when computing the ALS height statistics (*lascanopy* parameters: height_cutoff = 2, cover_cutoff = 2) [32] (Table 1).

**Table 1.** Set of statistics derived from ALS data computed for the ground-footprint location of GEDI laser beams across the training area.

| Variables | Description |
|---|---|
| **Height metrics: (height_cutoff = 2)** | |
| $h_{mean}$ | mean |
| $qav$ | quadratic mean height |
| $h_{std}$ | standard deviation |
| $h_{max}, h_{min}$ | maximum and minimum |
| $h_{Skw}$ | skewness |
| $h_{Kurt}$ | kurtosis |
| $CRR$ | canopy relief ratio ((mean heightmin height)/(max height- min height)) |
| $p_{01}, p_{10}, \ldots \ldots p_{99}$ | 5th, 10th, 20th, 25th, 30th, 40th, 50th, 60th, 70th, 75th, 80th, 90th, 95th, 99th percentiles |
| **Canopy cover metrics (cover_cutoff: 2 m)** | |
| (Canopy Cover) $CC_{ALS}$ | percentage of first returns above 2.00/total first returns |
| $PARA2$ | percentage of all returns above 2.00/total all returns |

### 2.3. GEDI Data Adquisition and Processing

GEDI-derived Level 2A (L2A) and Level 2B (L2B) data were used in this study (Table 2). GEDI L2A data contain the latitude, longitude, elevation, canopy height, and surface energy metrics ($rh0$, $rh10$, ... , $rh90$, $rh95$, $rh98$, $rh99$, $rh100$) extracted from return waveforms of the various reflecting surfaces located within each laser footprint [33]. The GEDI L2B standard data product adds vertical profile metrics: the canopy cover ($CC_{GEDI}$), plant area index ($PAI$), estimated vertical canopy directional gap probability for the selected L2A algorithm ($PGP\_THT$), and foliage height diversity index ($FHD$) for each laser footprint located on the land surface [34]. The bounding box of the Badajoz province was used to select all the GEDI beams within study area. There were 99 GEDI orbit tracks in HDF5 format available to the 15 July 2020 date (Figure 1). The GEDI laser shots considered in this research were collected in 2019 between 21 April and 31 October. The attribute Quality Flag (QF) was used to disregard all GEDI shots classified as 0, meaning that the technical and quality attributes for a given shot number (identifier) were not according to standards. A QF value of 1 indicates that a given shot number meets quality criteria based on energy, sensitivity, amplitude, and real-time surface tracking, and therefore, these shots were used for further analysis. See details of interpretation of L2A and L2B QF in [33,34]. The *rGEDI* package [35] was used to retrieve and process the GEDI data under the 3.6 version of the R statistical software [36] (R Core Team 2020).

**Table 2.** Set of statistics derived from GEDI Level 2A (L2A) and Level 2B (L2B) data computed for the ground-footprint location of GEDI laser beams across the training area.

| **(A)** GEDI Level 2A product | | | |
|---|---|---|---|
| **Label** | **Variable GEDI-AGB Model** | **Unit score** | **Description** |
| $rh$ | $rh01$, $rh02$, ... ... $rh100$ | m | Relative height metrics at 1% interval (m) |
| **(B)** GEDI Level 2B product | | | |
| *cover* | $CC_{GEDI}$ | % | Total canopy cover, defined as the percent of the ground covered by the vertical projection of canopy material |
| *pgap_theta* | $PGP\_THT$ | % | Canopy Gap Probability |
| *pai* | $PAI$ | $m^2/m^2$ | Total Plant Area Index |
| *fhd_normal* | $FHD$ | - | Foliage Height Diversity index calculated by vertical foliage profile normalized by total plant area index [37] |

The benchmark between ALS and GEDI was carried out within the boundaries of SFM polygons intersecting GEDI ground tracks. First, we selected GEDI shots completely contained within the forest type-specific SFM polygons (*Dehesas*, *Encinares*, *Alcornocales*,

*Pinaster,* and *Pinea*). Second, GEDI shots located further than 30 m from the edge of SFM boundaries were selected [27]. The final filter disregarded GEDI shots above the percentile 99th values observed SNFI-4 plots in Badajoz. The total set comprised 63,135 shots: 38,983 for *Dehesas*, 15,958 for *Encinares*, 3026 for *Alcornocales*, 1534 for *Pinaster*, and 3634 for *Pinea*. Finally, the matching between the GEDI- and ALS-derived canopy height metrics was evaluated in terms of Pearson's correlation coefficient (*r*) (Equation (1)), the overall root mean square error (RMSE, Equation (2)), the relative root mean square error (rRMSE, Equation (3)), Bias (Equation (4)), and rBias % (Equation (5)).

$$r = \frac{\sum_{i=1}^{n}(x_i - \overline{x}) \cdot (y_i - \overline{y})}{\sqrt{\sum_{i=1}^{n}(x_i - \overline{x})^2} \cdot \sqrt{\sum_{i=1}^{n}(y_i - \overline{y})^2}} \tag{1}$$

$$\text{RMSE} = \sqrt{\frac{\sum_{i=1}^{n}(x_i - y_i)^2}{n}} \tag{2}$$

$$\text{rRMSE} = \frac{\text{RMSE}}{\overline{x}} \times 100 \tag{3}$$

$$\text{Bias} = \frac{\sum_{i=1}^{n}(y_i - x_i)}{n} \tag{4}$$

$$\text{rBias} = \frac{\text{RMSE}}{\overline{x}} \times 100 \tag{5}$$

where $n$ is the number of GEDI shots, $x_i$ is the elevation ALS metric in m for the 25-m diameter footprint GEDI shot $i$, $y_i$ is the elevation estimation metric from the 25-m footprint GEDI in the GEDI Level 2A, $\overline{x}$ is the mean elevation observed values for the ALS-estimated at footprint GEDI level and $\overline{y}$ is the mean elevation observed values for the GEDI-estimated metric at footprint GEDI level.

### 2.4. Field Data Adquisition

The field data used for this study were obtained from the SNFI-4 in Extremadura. A total of 508 georeferenced plots with high-end positioning equipment were used to calibrate forest type-specific ALS-derived AGB models (Table 3). The field measurements of the SNFI-4 campaign in Extremadura were carried out during the year 2017. The uncertainty in the co-registration between ALS and field measurements was mitigated using high-performance global navigation satellite systems (GNSS) to improve positioning information. A handheld data collection system (TRIMBLE Juno 5B handheld, Trimble Inc., Sunnyvale, CA, USA) was used to determine the coordinates (error range 1–2 m of positioning error after post-processing) during field measurements. For further details of the procedure conducted to obtain the field data, see the protocol outlined in Álvarez-González et al., 2014 [6].

**Table 3.** Summary of ground data collected in the 4th National Forest Inventory (SNFI-4) for the five forest ecosystems. Plot-level estimates are presented for aboveground biomass (AGB, Mg ha$^{-1}$), stand basal area (*G*, m$^2$ ha$^{-1}$), and tree density (*N*, trees ha$^{-1}$).

| Forest Ecosystem | SNFI-4 Samples | Min AGB | Max AGB | Mean AGB | Min G | Max G | Mean G | Min N | Max N | Mean N |
|---|---|---|---|---|---|---|---|---|---|---|
| *Dehesas* | 239 | 4.11 | 154.36 | 41.20 | 1.13 | 19.50 | 6.17 | 5.09 | 969.08 | 86.37 |
| *Encinares* | 90 | 1.72 | 101.56 | 28.25 | 0.43 | 17.80 | 5.32 | 5.09 | 1310.16 | 284.88 |
| *Pinaster* | 82 | 1.80 | 184.48 | 73.95 | 0.59 | 46.46 | 20.51 | 14.15 | 1464.23 | 348.38 |
| *Alcornocales* | 45 | 1.69 | 112.41 | 29.85 | 0.54 | 25.64 | 8.26 | 10.19 | 1457.15 | 222.21 |
| *Pinea* | 52 | 11.07 | 159.90 | 49.46 | 2.77 | 39.88 | 12.41 | 29.43 | 1973.52 | 310.88 |

### 2.5. ALS-Derived AGB Models

We used (multiplicative) power–function models to establish empirical relationships between field measurements and ALS variables. The respective general expressions are as follows (Equation (6)):

$$y = \beta_0 \cdot X_1^b \cdot X_2^c \cdot \ldots \cdot X_n^m + \varepsilon \tag{6}$$

where $y$ is the estimated AGB from ALS; $X_1 \cdot X_2 \ldots \cdot X_n$ are potential explanatory ALS-derived variables related to metrics of height distributions or measurements related to canopy density (Table 1); $a$, $b$, $c$ are the parameters to be estimated by non-linear regression analysis; and $\varepsilon$ is the additive random error. The models were fitted using the *nls* function implemented in the *BASE* package of the R software [36].

Forest type-specific ALS-based AGB models for *Dehesas, Encinares, Alcornocales, Pinaster,* and *Pinea* were calibrated. The first step in the modeling phase was to select the optimal set of predictor variables to be used in the estimation of AGB. The *leaps* package, which is available for R [38], was used to select the significant predictors of the regression. In this study, we proposed the use of two predictors to estimate the parameter from the models. Collinearity between regressors was prevented by checking the variance inflation factor (VIF). In this study, regressors with VIF above 10 were disregarded [39]. In addition, a leave-one-out cross-validation (LOOCV) was performed for each potential regression model using programming routines in R [36].

Finally, we computed the model efficiency (Mef, Equation (7)), the overall root mean square error (RMSE, Equation (8)), the relative root mean square error (rRMSE, Equation (9)), the Bias (Equation (10)), and rBias (Equation (11)) to determine the accuracy of ALS-derived models for estimating AGB using four different modeling approach.

$$\text{Mef} = 1 - \left( \frac{(n-1) \sum_{i=1}^{n} (y_i - \hat{y}_i)^2}{(n-p) \sum_{i=1}^{n} (y_i - \overline{y}_i)^2} \right) \tag{7}$$

$$\text{RMSE} = \sqrt{\frac{\sum_{i=1}^{n} (y_i - \hat{y}_i)^2}{n}} \tag{8}$$

$$\text{rRMSE} = \frac{\text{RMSE}}{\overline{y}} \times 100 \tag{9}$$

$$\text{Bias} = \frac{\sum_{i=1}^{n} (\hat{y}_i - y_i)}{n} \tag{10}$$

$$\text{rBias} = \frac{\text{Bias}}{\overline{y}} \times 100 \tag{11}$$

where $n$ is the number of plots, $y_i$ is the field-estimated AGB in the plots $i$, $\overline{y}$ is the mean observed value for the field-estimated AGB in the plot, $\hat{y}_i$ is the estimated value of AGB derived from the non-linear regression model, and $p$ is the number of parameters in the models.

### 2.6. GEDI-Derived AGB Models

Firstly, the set of ALS-derived biomass models were applied to estimate AGB for the different forest ecosystems (*Dehesas, Encinares, Alcornocales, Pinaster,* and *Pinea* forests) at laser footprint level (~25 m), i.e., by using the ALS-derived metrics extracted from the extent of the GEDI shots. Secondly, the forest type-specific ALS-derived AGB estimates at footprint level was used as independent variable to develop GEDI-derived AGB models for each forest type, by using the upper *rh* metrics (*rh*60, *rh*70, *rh*80, *rh*90, *rh*95, *rh*98, *rh*99) from L2A and canopy profile metrics $CC_{\text{GEDI}}$, *PAI*, *PGP_THT,* and *LHD* from L2B as explanatories variables (Table 2). For GEDI-derived AGB models, the method was similar to ALS-derived AGB models. The empirical relationship between ALS characteristics and stand-level forest biomass suggests that common models based on metrics of height distributions or a combination between metrics of height distributions and measurements related to

the vertical canopy structure may be widely applicable to diverse forest types [40–42]. In addition, the combination of upper metric *rh*90, *rh*95, *rh*98, *rh*99, *CC*$_{GEDI}$, *PAI*, *PGP_THT*, and *LHD* was also tested. Then, we computed the performance of these forest type-specific models at GEDI footprint level in terms of Mef, RMSE, rRMSE, bias, and rBias as described in Section 2.5. Models were compared using Mef, RMSE, rbias, and a graphical inspection of the model residuals at the end of each model procedure. A leave-one-out cross-validation (LOOCV) was performed for each potential regression model using the R software [36].

## 3. Results

### 3.1. GEDI-ALS Metrics Accuracy

The relationship between *p*98 and *rh*98 was the best in terms of *r* correlation for the five forest ecosystems, except for *Dehesas* where *p*99–*rh*99 was slightly better in *r* (Table 4, Figures 2 and 3). The *p*98–*rh*98 comparison for 5 forest ecosystems yielded *r* Pearson values ranging from 0.49 to 0.65 for *Dehesas*, *Encinares*, and *Alcornocales*, and 0.71 for *Pinaster* and *Pinea* forests. In terms of rRMSE values, the error was slightly lower for the comparison between *p*98 and *rh*98 than for *p*95–*rh*95 and *p*99–*rh*99, except for *Dehesas* and *Pinaster*. The RMSE values of *p*98–*rh*98 for *Dehesas, Encinares, Alcornocales, Pinaster, and Pinea* were 2.05, 2.17, 1.95, 3.96, and 2.37 m, respectively; the rRMSE values were 29.39%, 38.68%, 31.14%, 28.63%, and 28.29%, respectively; while the bias values were −0.50, 0.39, −0.06, −0.97, and 0.27 m, respectively. In terms of bias and bias%, the *p*99–*rh*99 relationships were slightly better than *p*98–*rh*98 for *Dehesas* and *Pinaster*. Finally, Figures 2 and 3 depict the mean difference between *rh*98 and *p*98 metrics when the values are classified based on *CC*$_{ALS}$ for all the forest ecosystems. For the *Dehesas, Alcornocales,* and *Pinaster* formations, we found negative differences, on average, between *rh*98 and *p*98 when canopy cover is <50% (Figure 2b,f and Figure 3b), indicating that GEDI underestimates the canopy height (*rh*98) when compared with ALS *p*98. On the opposite, a mean positive difference between *rh*98 and *p*98 was observed for both *Encinares* and *Pinea* forest types, indicating that the GEDI canopy heights were higher than the ALS data across all the canopy cover classes, except when canopy cover is >90% for which the mean differences turn negative (Figures 2d and 3d). This could indicate GEDI limitations in penetrating dense canopy cover conditions, such as the case of *Encinares* and *Pinea* forest types.

**Table 4.** Comparison between the 95th, 98th, and 99th percentile ALS-based forest height (*p*) distribution and GEDI relative height (*rh*) metrics in terms of *r*, RMSE, rRMSE, Bias, and rBias.

| Forest Ecosystem | Metrics Comparison | Pearson Correlation (r) | Root-Mean-Square Error (RMSE, m) | Relative Root-Mean-Square Error (rRMSE, %) | Bias (m) | rBias (%) |
|---|---|---|---|---|---|---|
| *Dehesas* | *p*95–*rh*95 | 0.465 | 2.39 | 35.45 | −1.37 | −20.35 |
| | *p*98–*rh*98 | 0.496 | 2.05 | 29.39 | −0.51 | −7.26 |
| | *p*99–*rh*99 | 0.497 | 2.02 | 28.40 | −0.05 | −0.70 |
| *Encinares* | *p*95–*rh*95 | 0.529 | 2.03 | 38.26 | 0.40 | 7.52 |
| | *p*98–*rh*98 | 0.544 | 2.17 | 38.68 | 0.39 | 7.016 |
| | *p*99–*rh*99 | 0.545 | 2.36 | 41.37 | 0.82 | 14.46 |
| *Alcornocales* | *p*95–*rh*95 | 0.640 | 2.03 | 33.98 | −0.80 | −13.45 |
| | *p*98–*rh*98 | 0.651 | 1.95 | 31.14 | −0.06 | −0.99 |
| | *p*99–*rh*99 | 0.653 | 2.04 | 31.87 | 0.35 | 5.53 |
| *Pinaster* | *p*95–*rh*95 | 0.713 | 4.17 | 31.30 | −1.69 | −12.71 |
| | *p*98–*rh*98 | 0.716 | 3.96 | 28.36 | −0.96 | −6.86 |
| | *p*99–*rh*99 | 0.712 | 3.95 | 27.68 | −0.65 | −4.58 |
| *Pinea* | *p*95–*rh*95 | 0.718 | 2.36 | 29.80 | −0.53 | −6.76 |
| | *p*98–*rh*98 | 0.716 | 2.37 | 28.29 | 0.28 | 3.39 |
| | *p*99–*rh*99 | 0.709 | 2.51 | 30.05 | 0.70 | 8.41 |

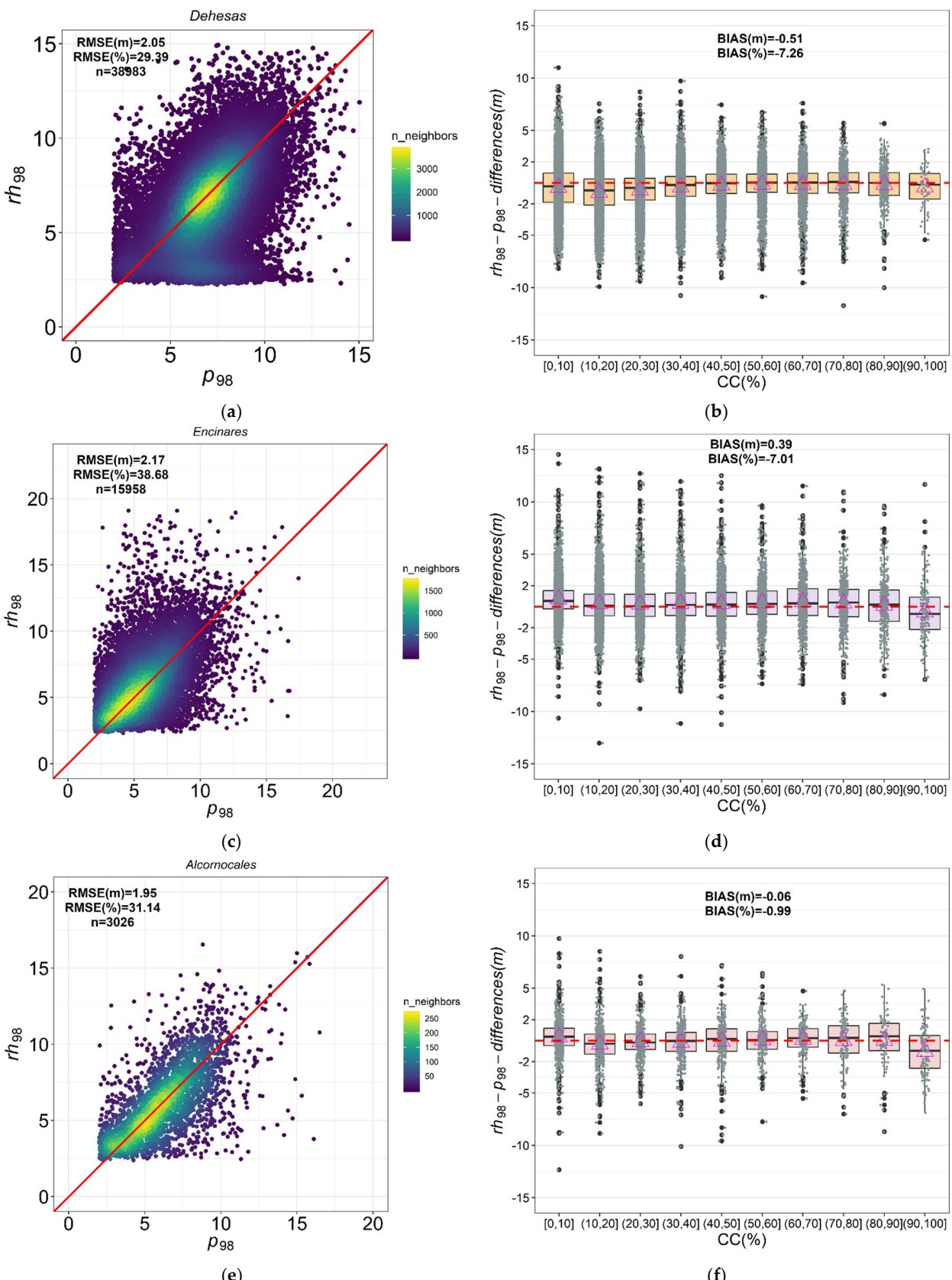

**Figure 2.** Scatterplots between ALS and GEDI-derived metrics for *p98rh98* and mean difference between ALS and GEDI metrics by $CC_{ALS}$ (%): (**a**,**b**) *Dehesas*; (**c**,**d**) *Encinares*, and (**e**,**f**) *Alcornocales*. The red solid line represents the 1:1 relationship in the scatterplots. Mean values of the differences between *rh98–p98* (triangle). The dashed red line represents y = 0 in boxplots.

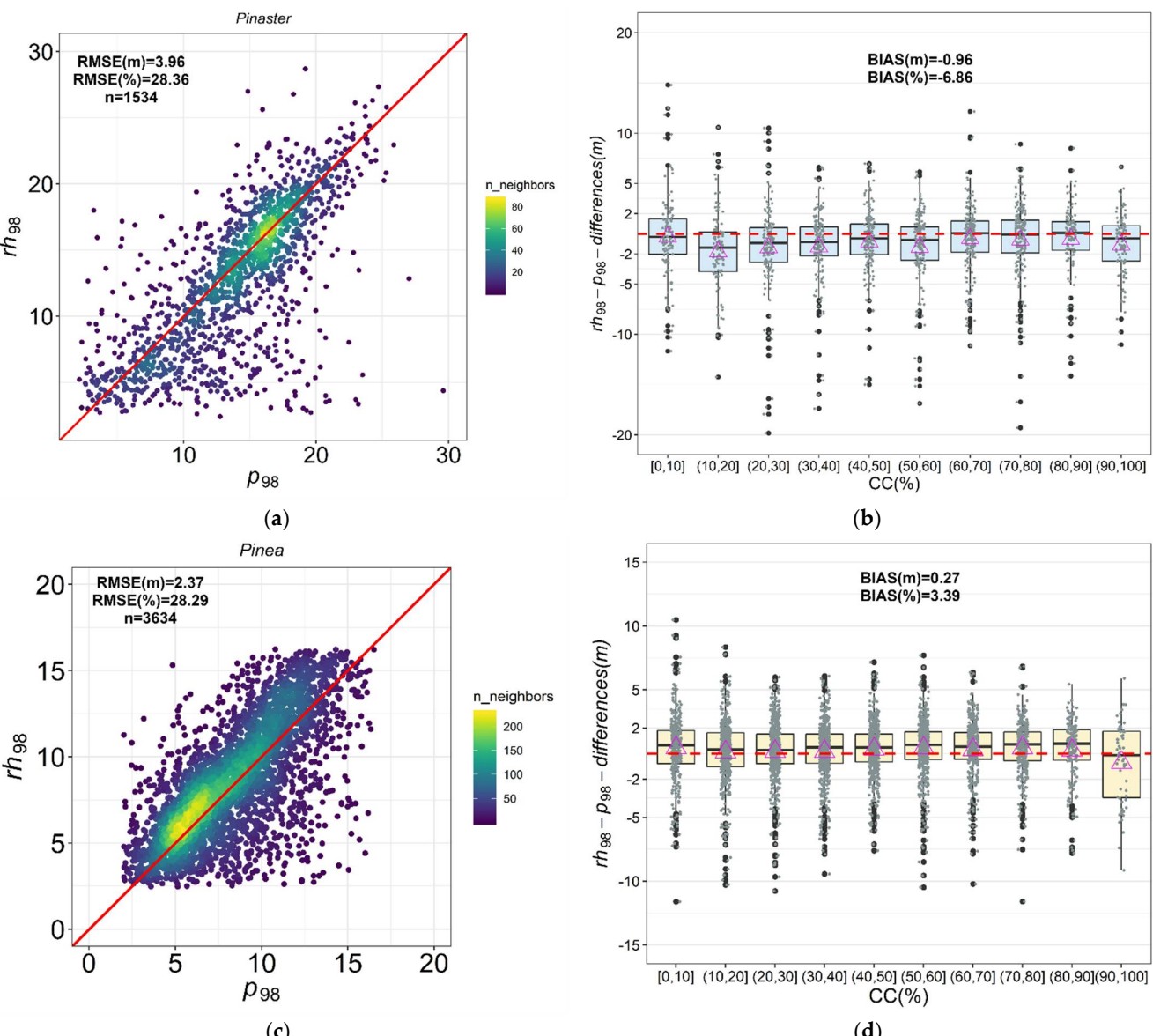

**Figure 3.** Scatterplots between ALS and GEDI-derived metrics for *p*98–*rh*98 and mean difference between ALS and GEDI metrics by $CC_{ALS}$ (%): (**a**,**b**) *Pinaster*; (**c**,**d**) *Pinea*. The red solid line represents the 1:1 relationship in the scatterplots. Mean values of the differences between *rh*98–*p*98 (triangle). The dashed red line represents y = 0 in boxplots.

### 3.2. ALS AGB-Derived Models

The performance of models for each forest type in terms of Mef, RMSE, and rRMSE are shown in Table 5. Non-linear regression models for AGB in *Dehesas*, *Encinares*, and *Alcornocales* forest ecosystems yielded Mef values ranging from 0.27 to 0.84 and from 0.76 to 0.86 for *Pinaster* and *Pinea*, respectively. In terms of rRMSE values, the values were slightly higher in *Dehesas* and *Encinares* (49.75 and 51.48%) than in *Alcornocales* (31.01%), *Pinaster* (37.01%) and *Pinea* (27.22%). In general, ALS-based models were better in terms of Mef, RMSE, and rRMSE in more closed *Pinus* and *Alcornocales* forests than in *Encinares* and *Dehesas* characterized by more open canopies. Table A1 (Appendix A) shows AGB estimation accuracies from LOOCV procedures by applying the best ALS-based AGB model summarized by the Mediterranean formations. There was no appreciable bias from the models throughout the observed *AGB* range using the best ALS-derived AGB models.

**Table 5.** Summary of the ALS-based AGB prediction models and plot-level accuracy assessment obtained for *Dehesas*, *Encinares*, *Alcornocales*, *Pinaster*, and *Pinea* forest ecosystems.

| Forest Type | Model | a | b | c | Mef | RMSE (Mg/ha) | rRMSE (%) | Bias | rBias (%) |
|---|---|---|---|---|---|---|---|---|---|
| | | | | | | Regression | | | |
| *Dehesas* | $AGB = a \cdot h_{20}^{b} \cdot CC_{ALS}^{b}$ | 3.00842 *** | 0.6914 *** | 0.491 *** | 0.27 | 20.4 | 49.75 | 0.18 | 0.48 |
| *Encinares* | $AGB = a \cdot p_{70}^{b} \cdot CC_{ALS}^{b}{}^{c}$ | 0.4387 * | 1.4052 *** | 0.5234 *** | 0.61 | 14.54 | 51.48 | 0.22 | 0.78 |
| *Alconocales* | $AGB = a \cdot p_{40}^{b} \cdot CC_{ALS}^{c}$ | 0.09626 * | 0.6912 *** | 1.283 *** | 0.84 | 9.26 | 31.01 | −0.77 | −2.58 |
| *Pinaster* | $AGB = a \cdot p_{20}^{b} \cdot PARA2^{c}$ | 0.31035 * | 0.3316 *** | 1.258 *** | 0.76 | 23.78 | 37.01 | −0.48 | −1.04 |
| *Pinea* | $AGB = a \cdot p_{30}^{b} \cdot PARA2^{c}$ | 0.15928 * | 0.988 *** | 1.0759 *** | 0.86 | 13.46 | 27.22 | 1.04 | 1.41 |

Pr(> | t | ) $p$ = <0.0001 '***' <0.001 '**' <0.01 '*' <0.05.

### 3.3. Performance of GEDI AGB-Derived Models

Table 6 reports the model performance of the GEDI-derived AGB models for the five analyzed forest types. Scatterplots of ALS-observed vs. GEDI-estimated AGB at GEDI footprint level are shown in Figure 4 for the best forest type-specific model in terms of Mef. Table A2 (Appendix A) shows AGB estimation accuracies from LOOCV procedures by applying the best model summarized by the Mediterranean formations. The negative and positive mean values in bias (Mg/ha) and rBias (%) indicate that the GEDI-derived AGB estimates are systematically underestimating (*Dehesa*, *Pinaster*, and *Pinea*) or overestimating (*Encinares* and *Alcorncoles*) ALS-based AGB estimates. Non-linear regression models for five formations yielded Mef values ranging from 0.31 to 0.46. In terms of rRMSE values, the values were slightly lower with the *Dehesas* (38.17%) and *Encinares* (57.87%) than *Alcornocales* (84.74%), *Pinaster* (48.19%), and *Pinea* (63.97%), respectively. In general, based on the histograms results for the best model (Figure 4b–j), the models slightly underestimated AGB over lower and higher intervals, corresponding with low and high canopy cover conditions. The models fitted with the best combination of one upper metric and one measurement related to vertical canopy structure were less unbiased than the models using stepwise selection methods throughout the observed AGB–ALS range. Models fitted with the combination of *rh95*, *rh98*, *rh99*, $CC_{GEDI}$ and *FHD* variables proved to be the most accurate and less unbiased models for *Dehesas*, *Alcornocales*, *Pinaster*, and *Pinea*, respectively. Although *PAI* and *PGP_THT* were also significant variables in the models, the proportion of variation explained by the regressions was lower than the models fitted by upper metrics in combination with $CC_{GEDI}$ and *FHD* in AGB modeling, except for *Encinares*, where *PGP_THT* was included in the best model. The model for pure homogenous *Pinea* forest yielded the best performance throughout the observed AGB (Figure 4i,j).

**Table 6.** Assessment of each GEDI-based AGB model dataset calculated with respect to the reference ALS-based AGB model estimates at the level of GEDI footprint.

| Forest Type | Model | a | b | c | Mef | RMSE (Mg/ha) | rRMSE (%) | Bias (Mg/ha) | rBias (%) |
|---|---|---|---|---|---|---|---|---|---|
| | | | | | | Regression | | | |
| *Dehesas* | $AGB = a \cdot rh99^{b} \cdot CC_{GEDI}^{c}$ | 10.69188 *** | 0.55525 *** | 0.10726 *** | 0.30 | 15.38 | 38.17 | −0.08 | −0.20 |
| *Encinares* | $AGB = a \cdot rh90^{b} \cdot PGP\_THT^{c}$ | 5.29572 *** | 1.06131 *** | 0.41344 *** | 0.33 | 14.13 | 57.87 | 0.14 | 0.65 |
| *Alconocales* | $AGB = a \cdot rh90^{b} \cdot FHD^{c}$ | 5.8822 *** | 1.50235 *** | −1.0564 *** | 0.38 | 22.06 | 84.74 | 0.71 | 2.73 |
| *Pinaster* | $AGB = a \cdot rh98^{b} \cdot CC_{GEDI}^{c}$ | 21.21140 *** | 0.56900 *** | 0.20040 *** | 0.37 | 32.16 | 48.19 | −0.45 | −0.67 |
| *Pinea* | $AGB = a \cdot rh95^{b} \cdot CC_{GEDI}^{c}$ | 10.40710 *** | 0.90480 *** | 0.25550 *** | 0.46 | 28.37 | 63.97 | −0.56 | −1.27 |

Pr(> | t | ) $p$ = <0.0001 '***' <0.001 '**' <0.01 '*' <0.05.

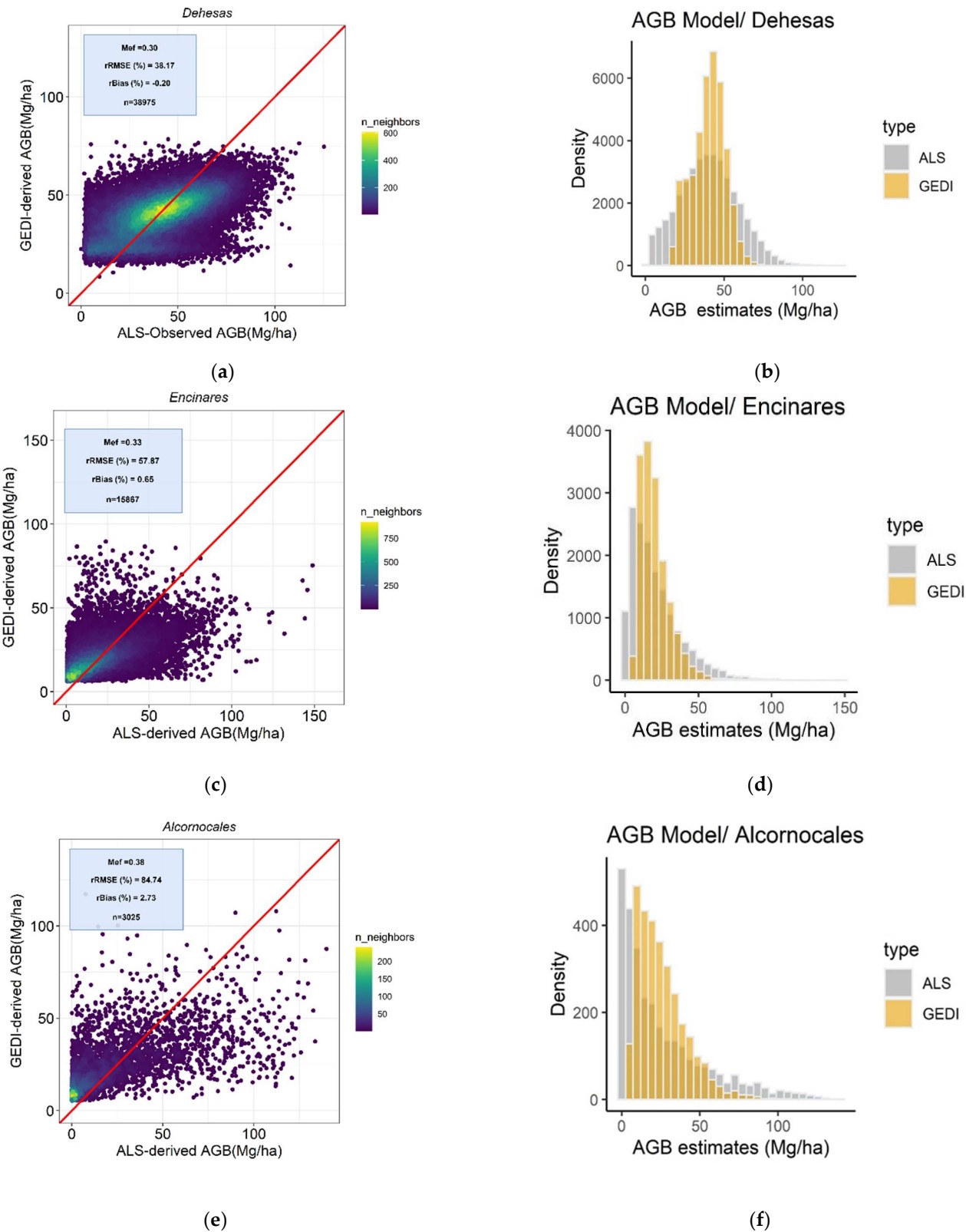

**Figure 4.** *Cont.*

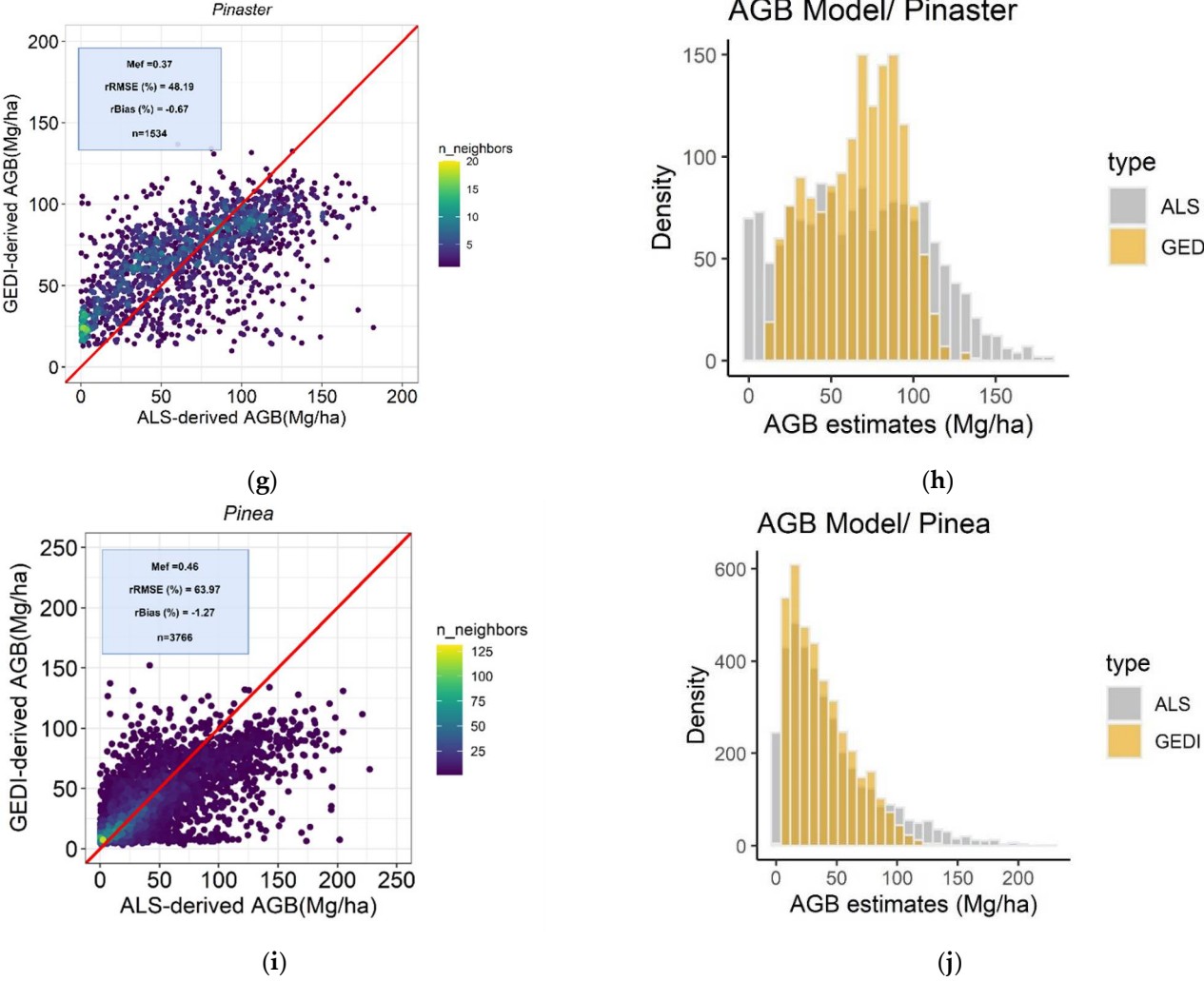

**Figure 4.** Scatterplots of ALS observed vs. GEDI-estimated at GEDI footprint level values of AGB for the best model in terms of rBias and associated histogram: (**a**,**b**) *Dehesas*; (**c**,**d**) *Encinares*; (**e**,**f**) *Alcornocales*; (**g**,**h**) *Pinaster*; (**i**,**j**) *Pinea*, canopy. The red solid line represents the 1:1 relationship.

## 4. Discussion

The usability of previously developed AGB models from ALS surveys and expensive fieldwork campaigns is especially relevant under region-wide sampling designs. The NASA GEDI mission brings an opportunity to update estimates of biomass across vast forest landscapes. The laser technology applied in FW GEDI differs from the discrete-return scanning carried out under mainstream ALS applications. Therefore, it is important to evaluate and understand the potential of GEDI data for its integrations in ALS-based workflows in forest inventory and forest management. Our study evaluated the ALS and GEDI performances using robust AGB models under the same temporal co-registration between ALS and GEDI datasets, and over five forest types with different complexities in terms of vertical and horizontal structure of the canopies. The bias in the prediction biomass estimates using GEDI observations as independent data depends on the model structure and the predictor variables included. Despite the fact that some recent studies in the literature assessed the accuracy of real [10,25] or ALS-simulated GEDI data [20,21,43,44], our study is, to the best of our knowledge, the first one to evaluate the performance of on-orbit GEDI L2A and L2B (Version 1) products in obtaining AGB estimates at footprint level (GEDI-like Level4A) by comparing with spatially and temporally coincident, discrete-return ALS data across vast areas of diverse Mediterranean types of forests.

Regarding the accuracy of GEDI canopy height estimates, our results are in accordance with the previously published studies that analyzed simulated GEDI data from pre-launch GEDI mission (e.g., Silva et al., 2018 [43] (p. 3517, $p98 = rh98$, RMSE = 2.99 m, bias = 0.47 m, bias(%) = 1.50, $n$ = 2987) and Hancock et al., 2019 [45] (p. 306, $p98 = rh98$, RMSE = 4.78 m, bias = 0.22 m). However, it is important to evaluate the accuracies of post launch GEDI data and products, since they might differ from the accuracies of the ALS-simulated GEDI data [15,36,38] due to possible geolocation inaccuracies or spatiotemporal variations in atmospheric attenuation [26]. The use of the 95th percentile resulted in more negative and positive bias and less accurate estimates than the 98th (RMSE increased from 2.03 m in *Alcornocales* to 4.17 m in *Pinaster*). In terms of bias and bias%, the $p99$–$rh99$ relationship was slightly better than $p98$–$rh98$ for *Dehesas* and *Pinaster*. The results confirm that the accuracy of GEDI–FW estimates of canopy height depends on the complexity of the horizontal and vertical structure of the Mediterranean vegetation. The GEDI footprint estimates were better in more closed and homogeneous coniferous forests of *P. pinaster* and *P.pinea* species ($r$ = 0.71) than in open canopy *Quercus*-dominated forests with values of $r$ ranging from 0.50 (sparse *Dehesas*) to 0.61 (multi-layered *Alcornocales*). The performance between metrics were similar in terms of RMSE and bias to recent published research at the European level [28] (RMSE = 2.5 m, rRMSE = 45%, and positive bias = 0.70), although our rRMSE values were slightly better from our study ranging from 28.29% to 38.68%. ALS–GEDI metrics accuracy for *Quercus*-dominated forest were similar in terms of $r$ to the results obtained by Adam et al., 2020 [25] who compared GEDI waveform ($rh100$) to the maximum of ALS-derived ($hmax$) canopy height model in two different temperate forests in central Germany ($r$ = 0.52–0.58, $R^2$ = 0.27–0.34, bias = −0.23–2.11 m). The relationship between $p98$ and $rh98$ in *Pinus*-dominated forests ($r$ = 0.71) was slightly lower than the one reported by Potapov et al., 2020 [10] when comparing $p90$ and $rh95$ ($r$ = 0.84, $R^2$ = 0.71, bias = −0.7 m, RMSE = 6.5 m, $n$ = 23,491). This was probably due to the low density of the ALS data used in our study (approx. 1 pulses m$^{-2}$) compared with that used by Potapov et al., 2020. However, our values in terms of bias (−0.97 for *Pinaster* and 0.28 for *Pinea*) and RMSE (RMSE = 3.8 m for *Pinaster* and RMSE = 2.37 m for *Pinea*) were better than the values obtained by Potapov et al., 2020. In any case, ALS reference data from Adam et al., 2020 and Potapov et al., 2020 might have also influenced their results, since there was a large time difference between ALS-acquisition dates (2012 and 2017) and GEDI-acquisition dates (2019). Mediterranean vegetation has, in general, low complexity if we compare it with complex vegetation structures such as tropical forests. This fact may be the main reason for the relatively low errors compared with other recent published studies [10,28]. Potapov et al., 2020 [10] compared GEDI relative height metrics data ($rh90$, $rh95$, and $rh100$) to the 90th percentile of ALS-derived height distribution ($p90$). The authors found that $rh90$ underestimated canopy height compared to $p90$ (mean difference −2.3 m) and $rh100$ overestimated it compared to $p90$ (mean difference +2.7 m). Our results showed that $rh95$ tends to underestimate canopy height compared to ALS-derived canopy height estimations for *Dehesas* (bias = −1.37), *Pinaster* (bias = −1.69), *Pinea* (bias = −0.53), and *Alcornocales* (bias = −0.80) and $rh99$ tends to overestimate canopy height compared to ALS-derived canopy height estimations in forest ecosystems as *Alcornocales* (bias = 0.80), *Encinares* (bias = 0.40), and *Pinea* (bias = 0.70). The results confirmed that GEDI height metric $rh95$ tends to underestimate canopy height when compared with ALS data in Mediterranean areas of sparse tree cover as reported by Potapov et al., 2020 [10]. Regarding the effects of canopy cover on the accuracy levels of GEDI height estimates, our results revealed that canopy heights were most accurate in the 50–90% range of canopy cover, and tend to present higher errors in dense cover (>90%) conditions as documented by [45] (see Figure 7d with $rh90$ in [45]). Neuenschwander et al., 2020 [29], in a study focused on assessing the accuracy of ICESat-2 data for canopy height estimates, also found this pattern. This confirms that, at low canopy cover (<50%) conditions, both ICESat-2 and GEDI full-waveform (FW) energies are more likely to be reflected from the terrain surface rather than the canopy, which precludes an accurate estimation of canopy height [33,34].

On the contrary, for the dense canopy cover conditions (CC > 90%) the terrain-reflected signal received by the GEDI and ICESat-2 sensors is more weak than the canopy signal leading to errors in canopy heights measurements [28,29,33].Therefore, *rh* metrics may be biased particularly in extreme (low and high) canopy cover conditions. The purpose of this study was to assess the performance and usefulness of the first release of the GEDI data (Version 1), which has a systematic geolocation error around 10–20 m [33,34]. As such, the accuracy of this data version was assessed without performing any geolocation error correction. Hence, the lower performance of the GEDI-derived canopy height in low density tree cover conditions, such as in *Dehesas*, can also reflect the impacts of the GEDI (Version 1) geolocation errors. In this type of ecosystem, the spatial fuzziness caused by the tree density variability can preclude the true comparison between GEDI measurements and the observed measurements on the ground. In a scattered tree ecosystem, such as *Dehesas*, an horizontal offset between 10 and 20 m can result in several meters of height errors, affecting model calibration and validation at the GEDI footprint level [10].

The GEDI mission was specifically designed to retrieve vegetation structure and AGB under a large range of environmental conditions sufficient to meet AGB mapping requirements. Regarding the exercise of developing GEDI AGB models using ALS AGB equations as similar to what GEDI's footprint level AGB product 4A will produce, the results suggested that existing ALS–AGB estimates could be used to generate robust GEDI-derived AGB models to predict AGB at footprint level in Mediterranean areas. For any spaceborne biomass estimate, validation using reference data is challenging, given that almost all reference data will have errors [19]. The results of the present study showed that GEDI-derived AGB models based on upper *rh* metric, $CC_{GEDI}$, *FHD,* and *PGP_THT* represent a sufficient quantitative description of Mediterranean structure analyzed at 25-m diameter footprint GEDI level using ALS-derived AGB estimates as reference. In terms of RMSE and rRMSE for the five forest ecosystem (RMSE = 15.25, 14.13, 22.06, 1.87, 27.95 Mg/ha, and rRMSE 37.85%, 57.87%, 84.74%, 47.75%, 63.02% for *Dehesas*, *Encinares*, *Alcornocales*, *Pinaster,* and *Pinea*, respectively), the precision of the GEDI-derived AGB models were similar or better (except in *Alcornocales* and *Pinea*) than those values reported by Duncanson et al., 2020 [20] (using GEDI simulations and a locally trained biomass model, calibrated against the ALS 30-m reference map in Sonoma County (US) (p. 111779-Table 2, rRMSE = 57.1%)). In *Dehesas*, *Encinares,* and *Pinaster* modeling, the rRMSE values achieved were also similar or better than to those reported by Silva et al., 2021 [21], who obtained an rRMSE of 54% for Sonoma County (US), with GEDI and ICESat-2 fused *AGB* calibration at a regular grid-based using GEDI's AGB models from Duncanson et al. 2020. However, the more complex and multilayered forest as *Alcornocales* (rRMSE = 84.74%) was the least accurately modeled of the Mediterranean forest. Our GEDI-derived AGB models using a combination of canopy height and vertical canopy structure metric from L2A and L2B product, respectively, were less unbiased in terms of bias and rbias (bias = −0.08, 0.14, 0.71, −0.45, −0.56 Mg/ha, rbias = −0.20%, 0.65%, 2.73%, −0.67%, and 1.27%, for *Dehesas*, *Encinares*, *Alcornocales*, *Pinaster,* and *Pinea*, respectively) than those values reported by Duncanson et al., 2020 [20] (bias = −26.3 Mg/ha and bias% = −18.7%) using US-wide GEDI AGB models based on only *rh* metrics and simulated AGB estimates at footprint level (GEDI-like Level4A as our study). In terms of rbias, the values were also slightly better than the values reported by Silva et al., 2021 [21] (bias% = −5.60%) using GEDI, ICESat-2, and NISAR fusion. In general, the GEDI-based AGB models were slightly negatively biased at lower and higher intervals, meaning that GEDI derived AGB models underestimated AGB under low and dense canopy cover conditions, as previous studies using simulated GEDI data [20]. There was no appreciable bias from the models throughout the observed AGB in pure homogenous *Pinea* forests (Figure 4i,j), in comparison with more complex vegetation as *Alcornocales* and *Encinares*. Our models yielded slightly lower values of Mef (0.31 to 0.46) (similar as adj. $R^2$) to those obtained by Silva et al., 2021 [21] (adj. $R^2$ ranging from 0.46 to 0.51). GEDI-derived models from our study captured AGB variations slightly worse, probably due to the following: (i) a wide variety of vegetation was analyzed

in AGB modeling from the same dataset by Duncanson et al., 2020, (ii) ALS reference map was used to calibrate instead of applying the trained model to ALS point cloud metrics derived at GEDI footprint level as in our study, (iii) Duncanson et al., 2020 and Silva et al., 2021 used ALS-simulated GEDI data and the ALS point cloud density was higher than in our study, and (iv) the influence of the GEDI (Version 1) geolocation errors into the models. If we compare with the ICESat2 mission, the values obtained in the present study were also worse, in terms rRMSE, than those reported by Narine et al., 2020 [46] using a simulated ICESat-2 vegetation product and ALS estimates AGB as reference (RMSE values were 28.90 Mg/ha or rRMSE = 37% of a mean value of 79 Mg/ha with the training dataset) in south Texas (US) (approximately 58% of the region or 80% of its forested area, predominantly coniferous forests). Our results demonstrated that metrics derived from L2A and L2B products at GEDI footprint level, such as canopy height (*rh*99, *rh*95, *rh*90) and vertical canopy structure metric ($CC_{\mathrm{GEDI}}$, *FHD*), were found to be significant explanatory variables for predicting AGB. We suggest that foliage height diversity index (*FHD*) [37], which measures the complexity of canopy structure, should be included as explanatory variables in *Alcornocales* formation. The use of canopy height metrics alone may omit some information in profiles with more vertical and horizontal heterogeneity, such as in natural Mediterranean forest. Conversely, vertical canopy structure metrics contributed to most models for estimating the AGB. The variable *PGP_THT*, based on the model from [47], in combination with *rh*90 upper metric, resulted in less biased models than GEDI-derived models using the variable canopy cover ($CC_{\mathrm{GEDI}}$) and plan area index (*PAI*) in *Encinares*. Our results also demonstrate that a specific second metric related to canopy cover ($CC_{\mathrm{GEDI}}$) from LiDAR waveforms is potentially useful for improving most of the models (Table 6). According to the results obtained, the set of models strengthened the idea that the combination of mean height and vertical canopy structure metrics represents a sufficient and concise quantitative description of a homogeneous vertical structure in Mediterranean Areas as *Pinea* forest. However, the results also showed the limitations of the GEDI spaceborne LiDAR mission in differentiating AGB and characterizing vegetation structure, specially under sparse canopy cover [22,29].

The trade-off between model accuracy when building AGB models and their use to predict AGB estimates using alternative data sources is an interesting hot research topic embraced under data fusion methods in forest monitoring and assessment. The result from this study would allow forest managers and scientists to better understand ABG dynamics in forest ecosystems, while adding value and use to existing ALS–AGB models developed in the past over valuable fieldwork data. The study from Adam et al., 2020 showed also the influence of low canopy conditions when validating GEDI estimates using multi-temporal ALS as benchmark. The assessment that we performed controlled the temporal co-registration and the spatial co-registration uncertainty, as our filters were less tolerant with the effect of edges and fragmentation. In the study by Adam et al., 2020, the use of ALS data collected in 2014, 2018, and 2019 somehow restricted the accuracy of the benchmark, as only part of the ALS surveys overlapped in time with the start of the GEDI mission. The integration of laser satellite products from the GEDI mission might be regarded as a first take before the launch of the BIOMASS satellite mission. Therefore, understanding the technology of satellite missions to address co-registration problems seems vital in order to combine the rich array of sources that forest managers have nowadays as national forest inventories, ALS campaigns, small-scale photogrammetry, and ongoing satellite image missions [17,29,48,49]. Upcoming version of GEDI products addressing geolocation could improve the results obtained in the Extremadura Region. In addition, further work is recommended to introduce the uncertainty of the ALS–AGB models and the forest species map were ignored in the modeling approach.

## 5. Conclusions

This study used real GEDI data for assessing canopy forest height and determining AGB in different Mediterranean forest ecosystems. Firstly, the results showed the accuracy

in canopy height from the L2A products using upper metrics from ALS. Secondly, GEDI calibration equations from Mediterranean forest were developed as a similar exercise to generate L4A products at footprint level (~25-m diameter). Our study highlighted the difficulty in differentiating AGB and characterizing vegetation structure under sparse forest cover, which is characteristic of Mediterranean forests. The findings provide an initial evaluation of the ability of GEDI to estimate AGB and serve as a basis for further upscaling efforts. For future challenges, reference canopy height, wall-to-wall ALS–AGB maps, and NFI plots from the Region of Extremadura will be used to validate the upcoming L4A and gridded Level 4B products where these footprints are used to produce mean AGB and its uncertainty in cells of 1 km. Further research could evaluate the robustness of the GEDI version 2 to compute part of the uncertainty in height/AGB caused by the mismatch in GEDI footprint geolocation and ALS point cloud data and how the new orbit corrections have improved the conditions to conduct ground-versus-ALS-versus-GEDI studies.

**Author Contributions:** Conceptualization, J.G.-H., A.P. and E.G.-F.; Methodology, I.D.-R., A.P., E.G.-F. and J.G.-H.; Data Analysis, I.D.-R., A.P. and J.G.-H.; Investigation, A.P., S.G., C.A.S., P.R.-G., E.G.-F. and J.G.-H.; Resources, J.G.-H.; Data curation, I.D.-R., A.P., B.B. and J.G.-H.; Writing—Original Draft J.G.-H., A.P., S.G., C.A.S., B.B., P.R.-G. and E.G.-F.; Preparation, A.P., J.G.-H., S.G., C.A.S., B.B., P.R.-G., E.G.-F. and J.G.-H.; Writing—Review & Editing, A.P., S.G., C.A.S., B.B., P.R.-G., E.G.-F. and J.G.-H.; Funding acquisition, J.G.-H.; Project Administration, J.G.-H.; Supervision, P.R.-G., E.G.-F., S.G., C.A.S., B.B. and J.G.-H. Coordination, J.G.-H.; J.G.-H. was involved as research coordinator and project manager in all the phases of the manuscript. All authors have read and agreed to the published version of the manuscript.

**Funding:** This work was partially supported by 'National Programme for the Promotion of Talent and Its Employability' of the Ministry of Economy, Industry, and Competitiveness (Torres-Quevedo program) via postdoctoral PTQ2018–010043 to Dr. Juan Guerra Hernández. This research was supported by the project "Extensión del cuarto inventario forestal nacional mediante técnicas LiDAR para la gestión sostenible de los montes de Extremadura" from the Extremadura Forest Service. The authors also thank to Forest Research Centre, a research unit funded by Fundação para a Ciência e a Tecnologia I.P. (FCT), Portugal (UIDB/00239/2020).

**Acknowledgments:** We gratefully acknowledge Vicente Sandoval and Elena Robla from the National Forest Inventory Department for supplying the inventory databases NFI data and the latest version of FMS.

**Conflicts of Interest:** The authors declare no conflict of interest.

## Appendix A

**Table A1.** Summary of absolute and relative RMSE for the calibrated ALS model and LOOCV AGB predictions stratified by vegetation; *n* = number of observations (field plots) per formation.

| Forest Type | *n* | Model | Cross One-Out Validation | | |
|---|---|---|---|---|---|
| | | | Mef$_c$ | RMSE (Mg/ha)$_C$ | rRMSE(%)$_C$ |
| *Dehesas* | 239 | $AGB = a \cdot h_{20}^b \cdot CC_{ALS}^b$ | 0.24 | 20.93 | 50.2 |
| *Encinares* | 90 | $AGB = a \cdot p_{70}^b \cdot CC_{ALS}^b{}^c$ | 0.54 | 15.32 | 55.87 |
| *Alconocales* | 82 | $AGB = a \cdot p_{40}^b \cdot CC_{ALS}^c$ | 0.80 | 10.39 | 38.58 |
| *Pinaster* | 45 | $AGB = a \cdot p_{20}^b \cdot PARA2^c$ | 0.69 | 27.37 | 36.71 |
| *Pinea* | 52 | $AGB = a \cdot p_{30}^b \cdot PARA2^c$ | 0.81 | 15.65 | 30.74 |

**Table A2.** Summary of absolute and relative RMSE for the GEDI calibrated model and LOOCV AGB predictions stratified by vegetation; *n* = number of observations per formation.

| Forest Type | n | Model | Cross One-Out Validation | | |
|---|---|---|---|---|---|
| | | | Mêf$_c$ | RMSE(Mg/ha)$_C$ | rRMSE(%)$_C$ |
| *Dehesas* | 38,983 | AGB $= a \cdot rh99^b \cdot CC^c_{\text{GEDI}}$ | 0.30 | 15.38 | 38.17 |
| *Encinares* | 15,958 | AGB $= a \cdot rh90^b \cdot PGP\_THT^c$ | 0.30 | 14.41 | 62.30 |
| *Alconocales* | 3026 | AGB $= a \cdot rh98^b \cdot CC^c_{\text{GEDI}}$ | 0.37 | 22.10 | 84.91 |
| *Pinaster* | 1534 | AGB $= a \cdot rh90^b \cdot FHD^c$ | 0.37 | 32.22 | 48.27 |
| *Pinea* | 3634 | AGB $= a \cdot rh95^b \cdot CC^c_{\text{GEDI}}$ | 0.45 | 28.41 | 64.07 |

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
