# Peer review of "Assessing the Accuracy of GEDI Data for Canopy Height and Aboveground Biomass Estimates in Mediterranean Forests"

_remotesensing, doi:10.3390/rs13122279_

Round 1

Reviewer 1 Report

This article shows an application of GEDI in the Mediterranean forests. The article is professionally written, simple to read and understand. The introduction shows a good state of the art and presents well the objective. The methodology and results are well exposed, and even is easy to replicate (thanks for that). Maybe the discussion is too long and there is a big quantity of data and acronymous that could be confused for the reader, simplify it could be welcome.  But overall, it is a good article and it has enough quality to be published in present form.

Author Response

Reviwer 1

General comment

This article shows an application of GEDI in the Mediterranean forests. The article is professionally written, simple to read and understand. The introduction shows a good state of the art and presents well the objective. The methodology and results are well exposed, and even is easy to replicate (thanks for that). Maybe the discussion is too long and there is a big quantity of data and acronymous that could be confused for the reader, simplify it could be welcome.  But overall, it is a good article and it has enough quality to be published in present form

Thank you for your positive comments

Reviewer 2 Report

The elimination of the word “transferability” and the explanation of the statistical processing without the use of this word is aright point to improve this new version of the manuscript. Authors also provide a better context for the complexity of the forest structure that they are analyzing against forest from other regions of the planet with higher complexity. They also used these differences in complexity to do a better comparison with the results of other scientists. In general, the minor issues also were well covered in this version of the manuscript; however, I need to highlight that authors still have problems with the acronyms, such as CCALS. This acronym still is not explained in the manuscript.

I recommend the publication of this manuscript in Remote sensing; however, authors need to review the acronyms and their definition in the manuscript. I do not need to review this issue. Main editor can review this minor problem before the publication of the paper.

Author Response

Reviewer 2

General comment

The elimination of the word “transferability” and the explanation of the statistical processing without the use of this word is aright point to improve this new version of the manuscript. Authors also provide a better context for the complexity of the forest structure that they are analyzing against forest from other regions of the planet with higher complexity. They also used these differences in complexity to do a better comparison with the results of other scientists. In general, the minor issues also were well covered in this version of the manuscript; however, I need to highlight that authors still have problems with the acronyms, such as CCALS. This acronym still is not explained in the manuscript.

Thank you for your positive comments. The definition of CCALS is now described in Table 1.

Reviewer 3 Report

The paper “ Assessing the accuracy of GEDI data for canopy height and  aboveground biomass estimates in Mediterranean forests” evaluates the accuracy and precision of GEDI data in estimating canopy height and AGB in a unique Mediterranean Ecosystem across five main species. ALS-derived height and AGB estimates were used to evaluated GEDI derived estimate. This paper identified the applicability and uncertainties of GEDI-derived height and AGB. The information could be useful for other studies when applying/choose GEDI data for large-scale mapping, particularly in similar ecosystems.

Here are detailed comments.

Table 1 the format is very difficult to understand, “Description”? Please clean and simplifier the table.

Figure 2 and Figure 3. the boxplot is not informative. Here are some suggestion: 1) add a y=0 line to show the positive/negative bias. 2. Use width of the box to indicate the sample size in each CC% types, 3. Remove the points and only use the 25%, 75% to emphasize the differenced among groups.

Is the location uncertainties of GEDI footprint considered? Will part of the final Height/AGB uncertainties be caused by the mismatch in GEID footprint geolocation and ALS point cloud data? How much is the uncertainty? A systematic evaluation of this uncertainty need to be quantified.

Author Response

Reviewer 3

General comment

The paper “Assessing the accuracy of GEDI data for canopy height and aboveground biomass estimates in Mediterranean forests” evaluates the accuracy and precision of GEDI data in estimating canopy height and AGB in a unique Mediterranean Ecosystem across five main species. ALS-derived height and AGB estimates were used to evaluated GEDI derived estimate. This paper identified the applicability and uncertainties of GEDI-derived height and AGB. The information could be useful for other studies when applying/choose GEDI data for large-scale mapping, particularly in similar ecosystems.

Here are detailed comments.

Table 1 the format is very difficult to understand, “Description”? Please clean and simplifier the table.

Thank you for your suggestion. A simplification of the table was done.

Figure 2 and Figure 3. the boxplot is not informative. Here are some suggestion: 1) add a y=0 line to show the positive/negative bias. 2. Use width of the box to indicate the sample size in each CC% types, 3. Remove the points and only use the 25%, 75% to emphasize the differenced among groups.

Thank you for the suggestion. We think it is important to show the positive/negative bias in class of 10% to detect clearly what happen across all the range. The dashed red line represents y=0 in boxplots was included in the Figure 2 and 3.

Is the location uncertainties of GEDI footprint considered? Will part of the final Height/AGB uncertainties be caused by the mismatch in GEID footprint geolocation and ALS point cloud data? How much is the uncertainty? A systematic evaluation of this uncertainty need to be quantified.

The purpose of this study was to assess the performance and usefulness of the first release of the GEDI data (Version 001). Future versions of GEDI data could be used to compute this uncertainty. We appreciate the comment and the idea to conduct successive tests. We were very selective with the filters aiming to fairly compared GEDI FW stats with discrete ALS statistics. We think our results are strongly supported with the filters, the high numbers of observations and geolocation error would not impact much as the forest structure is homogeneous within stands. Version 002 of the GEDI mission will shortly provided a better starting point for this kind of assessments as the nominal co-registration accuracy (i.e, XY error) is decreasing from the 20.9 to the 10.3 m range. The number of observations in this region-wide assessment is not a problem so using GEDI version 2 would provide more initial observations from which to compute tendency statistics at FME polygon level. Further research could evaluate the robustness of the GEDI version 002 towards data filterings and how the new orbit corrections have improved the conditions to conduct ground-versus-ALS-versus-GEDI studies.

This manuscript is a resubmission of an earlier submission. The following is a list of the peer review reports and author responses from that submission.

Round 1

Reviewer 1 Report

General comment

In this manuscript, authors sought to evaluate the effect of combining ALS-based aboveground biomass (AGB) models with GEDI-derived statistics by using temporally coincident datasets of discrete LiDAR and GEDI footprints. I have two main concerns in this manuscript. 1) Authors applied a method called temporal transferability that is cited but is not explained in the manuscript (authors must explain this method in the manuscript because temporal transferability does not have citations out of Spain). After reading the papers cited by the authors, temporal transferability seems like a correction for data sets with different temporality (for instance, LiDAR data and inventories surveys with different dates) using regressions. By applying transferability modeling, authors assumed that ALS-metrics (discrete LiDAR) and GEDI metrics (waveform LiDAR) can be coupled because they fit well. This assumption is not true. The problem with this assumption is that the point clouds of discrete LiDARs are not representative of the distribution of vertical surfaces of the vegetation as waveform LiDAR (specially in complex forests) because of the multiple scattering of LiDAR impulses (see this paper https://doi.org/10.1029/2018EA000506 ). Blair and Hofton (1999) developed a method to correct this issue that was set available to public by Hancock et. 2019 (the previous link). Authors must evaluate this method to improve the manuscript. Authors must write some lines and provide data of their analyses of this issue in the manuscript. 2) My second main concern is related, in some way, to the first main concern. Authors established a gradient of forest complexity for Mediterranean vegetation. Authors need to clarify in the manuscript that Mediterranean vegetation, in general, has low complexity (compare with real complex vegetation such as tropical forests). This could explain the relative low errors (compare with other published papers that author cited). But the most important, the low complexity of the Mediterranean vegetation could explain why the lack of a method to approach the discrete LiDAR to waveform LiDAR produced low errors. I have other smaller comments and concerts that authors can see below. I recommend that authors work to correct these issues and resubmit the manuscript to evaluate its possible publication in Remote Sensing.         

Other comments.

1) Pag 4, paragraph 2: Authors wrote “Finally, the above-ground height of ALS echoes was used to distinguish tree canopies (echoes above 2 m) and the shrub layer (echoes below 2 meters) when computing the ALS height statistics (lascanopy parameters: height_cutoff = 2, cover_cutoff = 2).”

Why two meters is the limit between trees and shrub layer? (could be 180 cm, 150 cm 250cm, etc). Author must provide the biological, ecological or other reasons to establish this limit (2m).

2) Pag6, paragrahp 1: Authors wrote: “where n is the number of GEDI shots, ?? is the elevation ALS metric in m and CCALS (%) for the 25-m diameter footprint GEDI shot i and ?? is the elevation estimation metric from the 25-m footprint GEDI in the GEDI Level 2A and CCGEDI (%) in GEDI Level 2B and ?Ì… is the mean observed values for the ALS-estimated at footprint GEDI level”

Authors starts to mention CC at this point of the manuscript (CCALS). Then, CC was added frequently in the methodology, results, and discussion. Although CC and CRR were added at the Table 1, it is not clear what exactly authors mean by CC and CRR. Authors must explain the exactly meaning of CC and CRR.

3) Pag7, paragraph 4: Authors wrote: “Finally, we computed the model efficiency (Mef, Eq 7), the overall root mean square error (RMSE, Eq. 8), the relative root mean square error (rRMSE, Eq. 9) and the Bias (Eq. 10) to determine the accuracy of ALS-derived models for estimating Wa using four different modeling approach”

Authors already added the equations for the errors measure RMSE, rRMSE, and BIAS previously. Authors must not repeat these equations in the manuscript. Authors must explain the use of these three errors measures (RMSE, rRMSE, and BIAS) briefly, avoiding the repetition of the equations.

4) Pag8, paragraph 2. Again, Authors already added the equations for the errors measure RMSE, rRMSE, and BIAS previously. Authors must explain the use of these three errors measures (RMSE, rRMSE, and BIAS) briefly, avoiding the repetition of the equations.

5) Pag7, paragraph 2. Authors have to provide more information about the ALS-derived modelling. The Table 3 seems to show that authors applied the models 1,2 and 3 to the total data while the model 4 was applied per vegetation type. If yes: why did authors apply the models in this way?   Did authors evaluate the effect of the vegetation as predictors variables? Authors must provide more information about these issues in the manuscript (or describe better these methods in order that readers can reproduce these analyses).  

6) The transferability model is a main topic in this manuscript, even is part of the title and objectives. Although transferability model was cited, transferability model must be described at some part of the manuscript maybe in the methods.

7) Figure 1f. Definitively, authors need to clarify the exactly meaning of CC. I assume that CC means pair of energy level maybe (RHs and percentiles-p). As well, authors need to clarify at methods what exactly mean the RHs of GEDI and p (percentiles of discrete LiDAR). What is the difference between both (rh are energy levels as well as p are energy levels or not?   Do rhs and percentiles fit well?   

I recommend to read

https://doi.org/10.1029/2018EA000506

https://doi.org/10.1016/j.rse.2020.112165

8) TABLE 5, 6 and 7. Could authors provide the standard deviations for Mef, RMSE, rRMSE, Mef, RMSE, rRMSEC? Maybe authors can run iterations of their calculations to estimate ranges values. I recommend this paper: https://doi.org/10.3390/rs11222697

9) Pag 15 first paragraph. Authors wrote: “Our study evaluated the ALS-versus-GEDI performance using robust biomass models under contemporary co-registration between ALS and GEDI, and over a set of forest ecosystems with different structural complexities in terms of vertical and horizontal structure of the canopies”.

Definitively, if authors state that their analyses are robust, they must run iteration of their calculations (Mef, RMSE, rRMSE, Mef, RMSE, rRMSEC) and provide the ranges by adding the standard deviations or other variation measure.

10) Pag 15, first paragraph. Authors wrote: “Despite some recent literature assessing the accuracy of real or simulated GEDI products by comparing with real or simulated ALS data (e.g., [10,25]).

Authors need to read and add this reference https://arxiv.org/abs/2103.03975

Reviewer 2 Report

This paper involved an examination of GEDI-derived statistics for estimating forest aboveground biomass (AGB) and an assessment of accuracy of canopy height and cover provided by GEDI. Non-linear regression models were used to estimate parameters, using four approaches to building the models.

Line numbers are missing from this manuscript, so a description of where changes are suggested are listed in the absence of specific page numbers. Line numbers are strongly recommended.

Page 2, paragraph 2: At the end of the first paragraph, a discussion of ICESat and relevant literature on vegetation studies over the region, with the data, should be included before introducing ICESat-2. The precursor to ICESat-2 provided essential EO data for enabling large coverage studies and thus should be mentioned.

Page 2, paragraph 2: Mapping estimations only at the country level? (larger spatial coverage anticipated?)

Page 2, paragraph 2: Did ICESat-2 developed representative pre-launch calibration equations for predicting AGB?

Page 3: Temporal resolution should be discussed here. Do GEDI and ICESat-2 offer repeat coverage to facilitate large-scale forest estimates on 3D forest structure in “almost real-time”?

Page 4, paragraph- How were the ALS data sets processed using LAStools? A citation is noted, but a brief description could be included as well.

Page 5: HDF5 instead of h5

Section 2.5: Was linear regression explored for modeling relationships? Why propose use of specifically two predictors?

Four modeling approaches specified; non-linear regression models noted, were other modeling techniques investigated?

The Results section contain many figures and tables, but briefly described. This section could be improved to describe the findings.

Are there scatterplots for the biomass models that may be included in the Results section?

Page 17: ICESat-2 instead of IceSat2